# Recognition of Continuous Face Occlusion Based on Block Permutation by Using Linear Regression Classification

**Jianxia Xue, Xiaojing Chen, Zhonghao Xie, Shujat Ali, Leiming Yuan** **, Xi Chen** **, Wen Shi \*** **and Guangzao Huang \***

College of Electrical and Electronic Engineering, Wenzhou University, Wenzhou 325035, China
\* Correspondence: shiwen@wzu.edu.cn (W.S.); guangzh@wzu.edu.cn (G.H.)

**Abstract:** Face occlusion is still a key issue in the study of face recognition. Continuous occlusion affects the overall features and contour structure of a face, which brings significant challenges to face recognition. In previous studies, although the Representation-Based Classification Method (RBCM) can better capture the differences in different categories of faces and accurately identify human face images with changes in light and facial expressions, it is easily affected by continuous occlusion. For face recognition, there is a situation where face error recognition occurs. The RBCM method frequently learns to cover the characteristics of face recognition and then handle face error recognition. Therefore, the elimination of occlusion information from the image is necessary to improve the robustness of such models. The Block Permutation Linear Regression Classification (BPLRC) method proposed in this paper includes image block permutation and Linear Regression Classification (LRC). The LRC algorithm belongs to the category of nearest subspace classification and uses the Euclidean distance as a metric to classify images. The LRC algorithm is based on one of the classification methods that is susceptible to outliers. Therefore, block permutation was used with the aim of establishing an image set that does not contain much occlusion information and constructing a robust linear regression model. The BPLRC method first modulates all the images and then lists the schemes that arrange all segments, enters the image features of various schemes into linear models, and classifies the result according to the minimum residual of the person's face image and reconstruction image. Compared to several state-of-the-art algorithms, the proposed method effectively solves the continuous occlusion problem for the Extended Yale B, ORL, and AR datasets. The proposed method recognizes the AR data concentration scarf to cover the accuracy of human face images to 93.67%. The dataset recognition speed is 0.094 s/piece. The arrangement method can be combined not only with the LRC algorithm, but also other algorithms with weak robustness. Due to the increase in the number of blocks and the increase in the calculation index of block arrangement methods, it is necessary to explore reasonable iteration methods in the future, quickly find the optimal or sub-best arrangement scheme, and reduce the calculation of the proposed method.

**Keywords:** face recognition; continuous occlusion; block permutation; linear regression classification

## 1. Introduction

With the emergence of the epidemic, face recognition technology has been rapidly implemented in various fields of people's daily life, such as in recognition of automatic passage on campuses, temperature measurement (all-in-one machines), and the use of face recognition in attendance. Traditional face recognition technology includes two main parts: feature extraction and classification. Gabor [1], the Histogram of Oriented Gradient (HOG) [2], and the Local Binary Pattern (LBP) [3] are often used to describe image features. Principal Component Analysis (PCA) [4,5] is based on the singular value decomposition (SVD) [6] algorithm. It performs eigen decomposition on the covariance matrix to obtain the principal components of the data and to achieve data dimensional reduction and extraction. The purpose of important features of PCA is also known as eigenfaces. The

feature extraction method, combined with the corresponding classification rules, has been used to solve initial face recognition problems. Common classifiers include the Nearest Neighbor Classifier (NNC) [7], the Minimum Distance Classifier (MDC) [8], the K-nearest Neighbor Classifier (KNNC) [9], and so on. Since traditional face recognition technology may overlook some important facial information or cause the over-fitting of problems in the feature extraction process, the Representation-Based Classification Method (RBCM) [10–13] has attracted much attention in recent years. Due to the epidemic's impact, people must wear masks while going out. The traditional face recognition method cannot effectively identify obscured faces, and the RBCM method is easily affected by abnormal features. In real life, face images often have problems such as illumination, facial expression changes, and facial occlusion. Among these problems, facial occlusion is considered the most challenging. From the literature [10–16], it can be seen that the RBCM method can effectively identify face images with changes in light and changes in facial expressions. Still, it is not easy to identify face images with occlusion. Therefore, the RBCM method is improved for facial occlusion to solve the three problems: light changes, facial expression changes, and facial occlusion.

To solve the problem of abnormal features in the RBCM, many research groups have proposed related robust algorithms, among which the Sparse Representation-based Classification (SRC) algorithm was among the first proposed [10,11]. The biggest feature of this type of algorithm is its ability to linearly represent the test samples by building a dictionary containing all the training samples. The relatively large time complexity of the SRC algorithm in solving the L1 norm optimization problem has largely limited its applications. However, Zhang et al. [14] proposed the Collaborative Representation-based Classification (CRC) algorithm. The SRC and CRC algorithms both use training samples from all categories to linearly represent test samples. However, the biggest difference between them is that the CRC algorithm uses the less computationally intensive L2 norm instead of the L1 norm, as used in the SRC algorithm, to solve the optimization problem. Some scholars have proposed a Two-Phase Test Sample Sparse Representation (TPTSSR) algorithm based on the CRC algorithm [15]. In the first stage of the algorithm, CRC was used to select M training samples for the best representation of test samples. In the second stage, the CRC algorithm was used again to identify the test samples, and the training set was the M training samples determined in the previous stage. The training samples selected in the first stage of TPTSSR cannot improve the accuracy of CRC for face image recognition. Thus, Tang et al. [16] further improved the algorithm and proposed the Random-filtering-based Sparse Representation (RFSR) algorithm. Liu et al. [17] improved the distance metric of the SRC algorithm. According to the author's experiments, using cosine or Euler distance as the measure can expand the inter-sample and intra-class distance simultaneously, and the multiple inter-class distance expansion was much higher than the multiple intra-class distance expansion, which was conducive to improving the robustness of the SRC algorithm. The various RBCM methods proposed in the literature [10–17] constrained the model coefficients to float in a small range, reducing the negative impact of face occlusion features on the model coefficients. Constraining the model coefficient through the L1 and L2 norm cannot improve model identification performance. Only when the model is constrained to learn as small a number of occlusion features as possible is model performance made stable.

In addition to using all classes of training samples to represent test samples linearly, representation-based classification methods can also use a single class of training samples to represent the test samples. Linear Regression Classification (LRC) [18] uses a single category of training samples to reconstruct the samples to be tested. LRC can be viewed as a representation based on the L2 norm, which uses the classification rules of the nearest subspace to classify face images. LRC finally selects the subspace with the smallest distance by projecting the test image onto the subspace. For this, the decision method is based on the distance metric, which is unsuitable for dealing with continuous severe occlusion problems. The LRC method is similar to the SRC and CRC methods, but the LRC method does not

restrict the representation coefficient. Therefore, if there are many abnormal variables in the training or test sample, the LRC method can learn a lot of abnormal information. As a result, the model classification fails. Therefore, some researchers, such as the authors of [18], have adopted the method of modular [19] image representation, which has made LRC more promising when trying to accurately identify face images with facial occlusions. The modular LRC algorithm [18] first worked by segmenting a given occlusion image. The "good or bad" image patch was then judged by the distance metric of the intermediate decision, and the "best" image patch was selected. Finally, converting the block into individual decisions was considered the final classification result. The main advantage of this method is its equivalency when dynamically removing occluded partitions. However, the biggest drawback of modular LRC is that it uses only specific blocks (with minimal residuals), discarding blocks that contain other useful face information. At this point, how to extract effective face characteristics and remove occlusion has become a question that researchers [20,21] care about.

This paper proposes the Block Permutation Linear Repression Classification (BPLRC) algorithm. The proposed method first modulates the image and then groups the schemes that retain the same number of blocks. Next, their residuals are compared to determine the permutation of the group and the group with the best recognition effect is finally selected. Compared to the modular LRC algorithm, the proposed method retains more useful face information and also achieves the purpose of removing invalid occlusion blocks. Moreover, there is no need to judge the occlusion ratio to achieve the best recognition effect. The algorithm improves the robustness of LRC to recognize the occluded image to a certain extent.

The RBCM methods used in this article mainly include LRC, SRC, CRC, Euler Sparse Representation-based Classification (ESRC [17]), Module LRC, and the BPLRC method proposed in this article. These methods are the same as linear models, and the type of linear representation is the difference. LRC uses a single class of travelers to reply to the test samples. SRC, ESRC, and CRC methods use all-class training samples to indicate test samples linearly. Among them, SRC and ESRC have differences in distance measurement. The difference between the SRC and CRC algorithm is that SRC uses the L1 model to restrain the sparse coefficient, and CRC uses the L2 model to restrain the sparse coefficient. Module LRC selects the "most useful" block as a feature input of the LRC algorithm through blocking. The difference from the LRC algorithm is that the Module LRC is used to screen the effective face characteristics as much as possible during identification, and the LRC algorithm simply uses all image features in identification. The BPLRC algorithm proposed in this article is similar to Module LRC. All images are modularized, and the identification of LRC algorithms characterizes effective face characteristics. The difference between the BPLRC algorithm and the Module LRC algorithm is that the Module LRC only retains one block, while the BPLRC method considers all the segmentation arrangement schemes to retain as much information as possible. From a principle point of view, all block arrangement schemes in BPLRC include the full-retained LRC and the Module LRC method that only retains one block.

The rest of this article is organized as follows: Section 2 details the principles of the LRC, SRC, CRC, and BPLRC algorithms, with a slight reference to ESRC and Module LRC. Section 3 describes the decision principles and experimental results of the proposed method. Section 4 discusses the contrast between BPLRC and other related algorithms in relation to the other performance metrics of the model. Section 5 is the conclusion of this paper.

## 2. Materials and Methods

The LRC [18] method can effectively identify face images with illumination and expression changes, but this method finds it challenging to identify occluded face images. According to the literature [14,17,20,21], LRC finds it easier to identify face images with illumination and expression changes than SRC, CRC, and ESRC algorithms in both linear models. Nevertheless, its robustness is weaker than the SRC, CRC, and ESRC algorithms.

The BPLRC algorithm proposed in this paper improves robustness based on the LRC algorithm to effectively identify face images with light changes, expression changes, and facial occlusion.

### 2.1. Related Works

Suppose that the training set samples are represented as $X = [X_1, X_2, \ldots, X_n]^T \in \Re^{n \times d}$, where $n$ is the number of categories contained in the training set samples and d is the dimension of any sample. Suppose the training sample of the $i$-th subspace is $X_i = [x_{i1}, x_{i2}, \ldots, x_{in_i}]^T \in \Re^{n_i \times d}$, denoted as $x_i \in \Re^d (i = 1, 2, \ldots, n)$, with $x_{ij}$ denoted as the $j$-th sample of the $i$-th class, $n_i$ denoted as the number of the $i$-th class of samples, and the test sample denoted as y.

The Linear Regression Classification algorithm is based on the assumption of subspace, and the data containing n categories are represented as n different subspace vectors, in which the samples belonging to the $i$-th category are represented as $X_i$. The following is the specific principle of the LRC [18] algorithm when classifying the test samples.

Assuming that the test sample y belongs to the $i$-th class, it can be approximately represented as a linear combination of the training samples of the same category.

$$y = X_i \beta_i, i = 1, 2, \ldots, n, \tag{1}$$

where $\beta_i$ is the representation coefficient of the $i$-th category of training samples.

Face recognition is expressed as a regression problem in the above formula, and the representation coefficient is obtained through the pseudo-inverse matrix.

$$\hat{\beta}_i = (X_i^T X_i)^{-1} X_i^T y, \tag{2}$$

where $y$ is the test sample vector.

The projection and projection surface of the test sample y in each subspace can be expressed as:

$$\hat{y}_i = X_i \hat{\beta}_i, i = 1, 2, \ldots, n, \tag{3}$$

$$\hat{y}_i = X_i (X_i^T X_i)^{-1} X_i^T y, \tag{4}$$

$$\hat{y}_i = P_i y, \tag{5}$$

where $\hat{\beta}_i$ is the predicted regression coefficient of the $i$-th category training sample, $\hat{y}_i$ is the linear model reconstruction vector, and $P$ is the projection matrix.

The distance between the test sample vector and the projection of y on the $i$-th subspace can be expressed as:

$$d_i(y) = \|y - \hat{y}_i\|_2, i = 1, 2, \ldots, n. \tag{6}$$

Selection of the category with the smallest Euclidean distance as the discrimination result obtains:

$$\text{identity}(y) = \underset{i}{\arg\min} \, d_i(y), i = 1, 2, \ldots, n. \tag{7}$$

### 2.2. Other Related Algorithms

Sparse Representation-based Classification (SRC) is based on linear regression through the punishment of regression coefficients. The SRC [10,11] algorithm introduces the L1 norm to constrain the regression coefficients so that more zero values are included in the regression coefficient, equivalent to using the Lasso regression model. The SRC algorithm first encodes the test samples as a sparse linear combination of all the training samples. It then makes the final decision by comparing which category has the smallest error.

The sparse coefficient of the SRC model can be equivalent according to the Lasso regression model:

$$\hat{\beta} = \underset{\beta}{\arg\min} \|y - X\beta\|_2^2 + \lambda \|\beta\|_1, \tag{8}$$

where $X$ is all the training samples, $y$ is the test sample, $\beta$ is the regression coefficient of $X$, and $\|\cdot\|_1$ is the L1 norm.

After determining the sparse coefficient of the SRC algorithm, the SRC algorithm is similar to the LRC algorithm, and the same equations as those of (3), (6), and (7) can use the Euclidean distance as a measure to determine the category of the test sample.

Since there is no analytical solution to the Lasso problem, its computational complexity is much greater than the classifier. Later, the L2 norm was introduced to restrict the regression coefficient, that is, the classification method (Collaborative Representation-based Classification, CRC).

The CRC [14] problem is equivalent to the ridge regression problem. It is similar to the SRC algorithm in that it similarly encodes the test samples as a linear combination of all the training samples. The CRC algorithm performs the L2 norm constraint on the regression coefficient and its sparse coefficient:

$$\hat{\beta} = \underset{\beta}{\arg\min} \|y - X\beta\|_2^2 + \lambda \|\beta\|_2^2, \tag{9}$$

where $X$ is all the training samples, $\beta$ is the regression coefficient of $X$, $\lambda$ is regularization coefficient, and $\|\cdot\|_2$ is the L2 norm.

The regression coefficients in Equation (9) have analytical solutions with the expression:

$$\hat{\beta} = (X^T X + \lambda I)^{-1} X^T y \tag{10}$$

By combining Equations (3), (6), and (7), the residuals of the CRC algorithm and the predicted test sample category can be expressed as:

$$d_i(y) = \frac{\|y - X_i \hat{\beta}_i\|_2}{\|\hat{\beta}_i\|_2}, i = 1, 2, \ldots, n. \tag{11}$$

$$\text{identity}(y) = \underset{i}{\arg\min} \, d_i(y), i = 1, 2, \ldots, n, \tag{12}$$

where $\hat{\beta}_i$ is the coefficient of the $i$-th category of training samples.

The SRC and CRC algorithms improve the linear regression classifier by restricting the L1 and L2 norm on the regression coefficient, respectively, which can somewhat suppress the influence of noise on the linear model. Nevertheless, the CRC algorithm is more computational than the SRC algorithm.

The Euler Sparse Representation-based Classification (ESRC [17]) algorithm is similar to the SRC algorithm, which takes Euler distance as a measure and expands intra-class and inter-class distance. The multiple of the inter-class distance will be greater than the multiple of the in-class distance in some data, thus improving the robustness of the SRC algorithm. The ESRC method involves an implementation process, and details of image mapping to the complex space process can found in [17].

### 2.3. The Proposed Method

The workflow of the BPLRC method is shown in Figure 1 under the assumption that the image is divided into 5 pieces. When the image only retains one block, the principle of the BPLRC algorithm is equivalent to Module LRC; when the image retains all blocks, the principle of the BPLRC algorithm is equivalent to LRC. The proposed method contains Module LRC that is reserved with a full block and blocks that only retain one block. Theoretically, its recognition effect will be better than LRC and Module LRC. BPLRC aims to find the best facial features for classification. The proposed method first divides all training images and a test image and divides the number of groups according to the number of reserved blocks. As shown in the figure, the number of arrangements from the first and fifth groups is 5, 10, 10, 5, and 1, respectively. The construction of a linear model according to each scheme then determines the scheme with the minimum residual and

obtains the identification result of the scheme. Finally, samples can be continually drawn from the test concentration and the above work can be repeated.

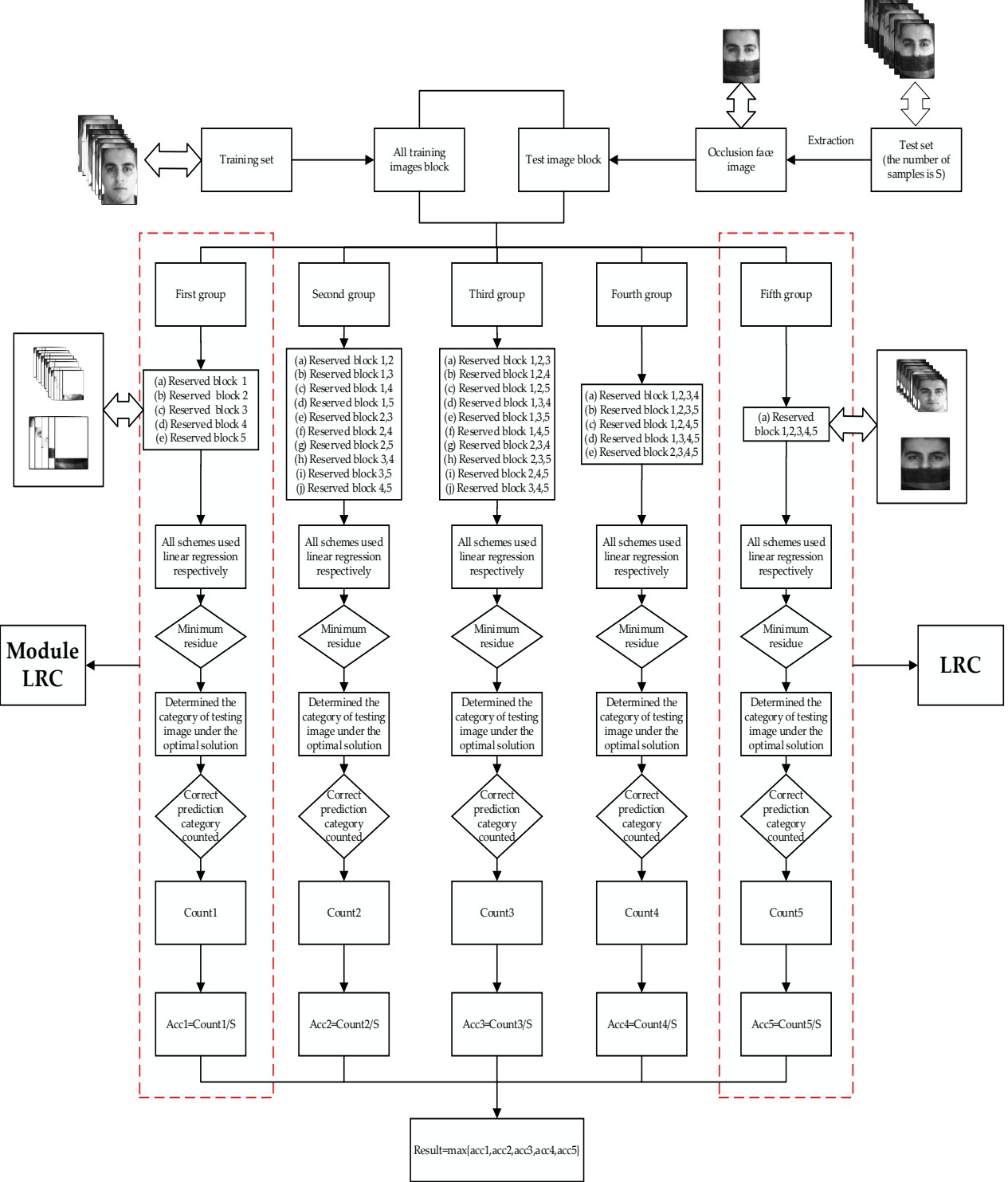

**Figure 1.** The overall workflow of the BPLRC method (with 5 blocks) [Reprinted with permission from Elsevier [20]. Copyright 2013, Neurocomputing].

Assuming that the training samples contain some noise when the test samples have the same noise, test samples can be linearly represented by the training samples. However, if the training samples contain no noise and the test samples contain a lot of noise, the

linear model will be invalid. Therefore, selecting the sample variables and filtering out the variables with noise in the samples are necessary, which can then help train a more robust linear regression model. The proposed method divides the variables into blocks and performs permutation, combination, and reorganization. The best combination method is then judged by Euclidean distance (residual value) to achieve the purpose of removing continuous noise variables. BPLRC aims to find face characteristics that contain only a small amount of noise or no noise. The following is the basic process of the BPLRC method:

Suppose the training samples on the *i*-th subspace are divided into *T* blocks such that each sub-image can be represented as:

$$U_i^{(t)} = [w_{i1}^{(t)}, w_{i2}^{(t)}, \ldots\ldots, w_{ip_i}^{(t)}], i = 1, 2, \ldots, n, \tag{13}$$

where $w_{ip_i}^{(t)}$ is the grayscale value on the $p_i$ pixel of the *t*-th block image from the *i*-th class.

The number of groups of arrangement schemes is determined by the number of divided blocks, and the block arrangement and combination of the *i*-th type of training samples in the first group can be expressed as:

$$\begin{cases} X_i^{(11)} = [U_i^{(1)}, O^{(2)}, O^{(3)}, O^{(4)}, \ldots, O^{(T)}]^T \\ \qquad\qquad \vdots \\ X_i^{(1m_1)} = [O^{(1)}, O^{(2)}, \ldots, U_i^{(m_1)}, \ldots, O^{(T)}]^T \end{cases}, m_1 = 1, 2, \ldots, C_T^1, \tag{14}$$

where *O* is a zero matrix and $U_i^{(m_1)}$ is the $m_1$-th block of class *i*.

Different groups retain a certain number of sub-images. For example, the second group retains 2 sub-images, and the *T*-th group retains *T* sub-images. Therefore, the 2-*T* groups *i*-th training sample reorganization can be expressed as:

$$X_i^{(2m_2)}, X_i^{(3m_3)}, \ldots, X_i^{(tm_t)}, t = 1, 2, \ldots, T; m_t = 1, 2, \ldots, C_T^t, \tag{15}$$

where $m_t$ is the $m_t$-th arrangement and $C_T^t$ is the number of arrangements.

Similar to Equations (14) and (15), the recombination test samples of different groups can be expressed as:

$$y^{(1m_1)}, y^{(2m_2)}, \ldots, y^{(tm_t)}, t = 1, 2, \ldots, T; m_t = 1, 2, \ldots, C_T^t. \tag{16}$$

where $m_t$ is the $m_t$-th arrangement and $C_T^t$ is the number of arrangements. Equations (14)–(16) represent different segmentation arrangements in the image. In this article, which refers to the method of Module LRC, the residues of all solutions are compared and the minimum residual arrangement scheme is found to obtain the best face characteristics used in identification. BPLRC reduces the continuous occlusion characteristics of linear model learning and increases the learning of effective face characteristics, thereby building a strong, robust linear model.

The coefficient vector and prediction vector for each group were calculated in the same manner as the expressions in Equations (2) and (3):

$$\hat{\beta}_i^{(tm_t)} = \left[ (X_i^{(tm_t)})^T X_i^{(tm_t)} \right]^{-1} (X_i^{(tm_t)})^T y^{(tm_t)}, \tag{17}$$

$$\hat{y}_i^{(tm_t)} = X_i^{(tm_t)} \hat{\beta}_i^{(tm_t)}, \tag{18}$$

where $X_i^{(tm_t)}$ is all the training samples of class *i* in the *t*-th group of scheme $m_t$.

In the same arrangement, the distance between the test vector and its projection on the *i*-th subspace is:

$$d_i(y^{(tm_t)}) = \left\| y^{(tm_t)} - \hat{y}_i^{(tm_t)} \right\|_2, i = 1, 2, \ldots, n, \tag{19}$$

where $y^{(tm_t)}$ is the test sample of the $m_t$ arrangement scheme in $t$-th group and $\hat{y}_i^{(tm_t)}$ is the prediction vectors of class $i$ in the $t$-th group of scheme $m_t$.

Selection of the distance to the nearest subspace of the test vector in the same arrangement:

$$d^{(tm_t)} = \min d_i(y^{(tm_t)}), i = 1, 2, \ldots, n, \tag{20}$$

where $d_i(y^{(tm_t)})$ is the distance between the test sample vector and the $i$-th subspace in the $t$-th group of scheme $m_t$.

Comparison of the size of the nearest subspace distance between different arrangements in the same group and selection of the arrangement with the smallest distance as the optimal arrangement in the group is expressed as:

$$D_t = \underset{m_t}{\arg\min} \, d^{(tm_t)}, m_t = 1, 2, \ldots, C_T^t, \tag{21}$$

where $d^{(tm_t)}$ is the distance (residual) of the $m_t$-th arrangement in $t$-th group.

Under the optimal scheme, the subspace closest to the recombination test sample is selected as the prediction result:

$$identity(y) = \underset{i}{\arg\min} \, d_i(y^{(tD_t)}), i = 1, 2, \ldots, n, \tag{22}$$

where $d_i(y^{(tD_t)})$ is the distance between the reorganization test sample vector and the $i$-th subspace under the optimal scheme in the $t$-th group.

Assuming that the number of test samples is S, the true label is $A^{ts_t} \in \{1, \ldots, C\}$ and the predicted label is $\hat{A}^{ts_t}$, the model's recognition accuracy can be expressed as:

$$acc^t = \frac{1}{S} \sum_{s_t=1}^{S} I(A^{ts_t} = \hat{A}^{ts_t}), t = 1, 2, \ldots, T, \tag{23}$$

where $I(\cdot)$ is the indicator function.

According to Equation (23), the recognition accuracy of each group was obtained, and the final result was obtained by comparison:

$$result = \max acc^t, t = 1, 2, \ldots, T, \tag{24}$$

where $acc^t$ is the accuracy of the identification of all test samples in the $t$-th group.

In addition to recognition accuracy, precision and recall can be partially evaluated to evaluate the model. Precision and recall can be expressed as:

$$\mathrm{Pr}ecision = \frac{TP}{TP + FP}, \tag{25}$$

$$\mathrm{Re}call = \frac{TP}{TP + FN}, \tag{26}$$

where $TP$ is actually a positive sample prediction as a positive sample, $FP$ is actually negative sample prediction as a positive sample, and $FN$ is actually positive sample prediction as a negative sample.

*F1-Score* can represent the harmonic average of accuracy and recall rate:

$$F1Score = \frac{2\mathrm{Pr}ecision \cdot \mathrm{Re}call}{\mathrm{Pr}ecision + \mathrm{Re}call}. \tag{27}$$

## 3. Results

### 3.1. Data Sources and Operating Environment

The effectiveness of the proposed method was demonstrated based on three standard databases, namely AR [22], Extended Yale B [23], and ORL [24]. These databases contain

several deviations from ideal conditions, including pose, lighting, occlusion, and gesture changes. Appropriate experimental results have demonstrated that the developed method performs well for severe continuous occlusion with small changes in pose, scale, illumination, and rotation. All the above experiments were run on the Windows 10 operating system (Intel Core i7-4770 CPU M620 @ 3.40 GHz and 8 GB RAM), and the programming environment was Python 3.7.

### 3.2. Selection of Optimal Block Arrangement and Combination Scheme

By taking the AR data as an example, the AR face database subset [22], comprising 50 males and 50 females, contains 2600 images in total. In the experiment, eight images without facial occlusion, such as smiling and not smiling, and brightness changes were selected as training samples for each object. Moreover, three face images covered by sunglasses and three face images covered by scarves were selected as test samples for each object. A total of 100 objects were selected. References [14,17,18,20,21,25] were selected for this experiment, and the images, with a resolution of 165 × 120 pixels, were downsampled to 15 × 10 pixels, 20 × 15 pixels, and 25 × 20 pixels for experiments.

The occluded face image contains face information and non-face information. Since the distribution of non-face information cannot be perceived in advance by algorithms, it is difficult to distinguish between face information and non-face information. Figure 2 shows a part of the occluded face images from the AR subset.

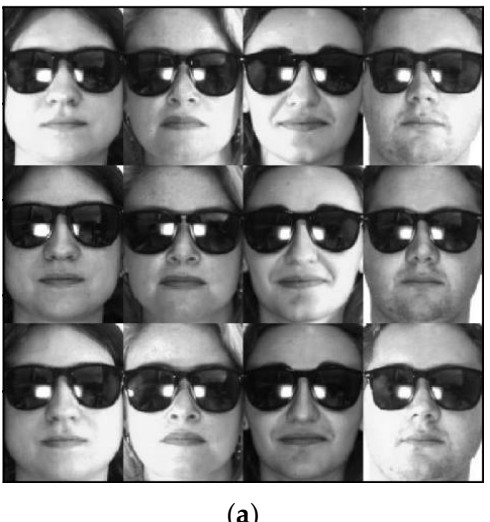 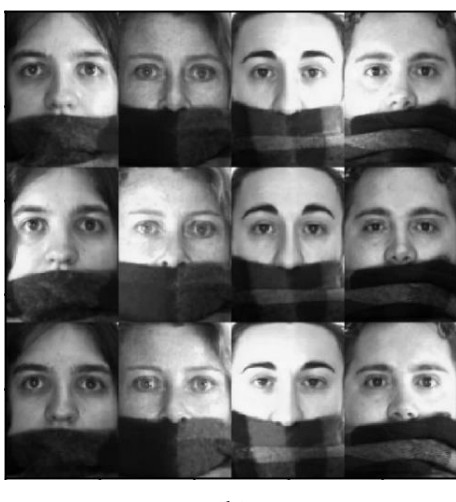

(**a**)          (**b**)

**Figure 2.** Faces from the AR subset occluded by sunglasses and scarves: (**a**) face images occluded by sunglasses; (**b**) face images occluded by scarves [Reprinted with permission from Elsevier [20]. Copyright 2013, Neurocomputing].

In this paper, the block arrangement method was adopted. All the blocks were arranged, combined, and finally compared to the residual values of each scheme to determine the final arrangement (see Figures 3–5). During this process, the approximate position of the obstructions, such as sunglasses or scarves, was determined. Considering the amount of computation, when taking three blocks in the five-block image as an example, the number of permutation schemes was 10. Only the horizontal block combination is shown in the figure, but the vertical block is also considered. The residual calculation of the arrangement of scheme h is shown in Figure 5, and Figure 6 shows the minimum residuals of face images involving sunglasses with different permutations.

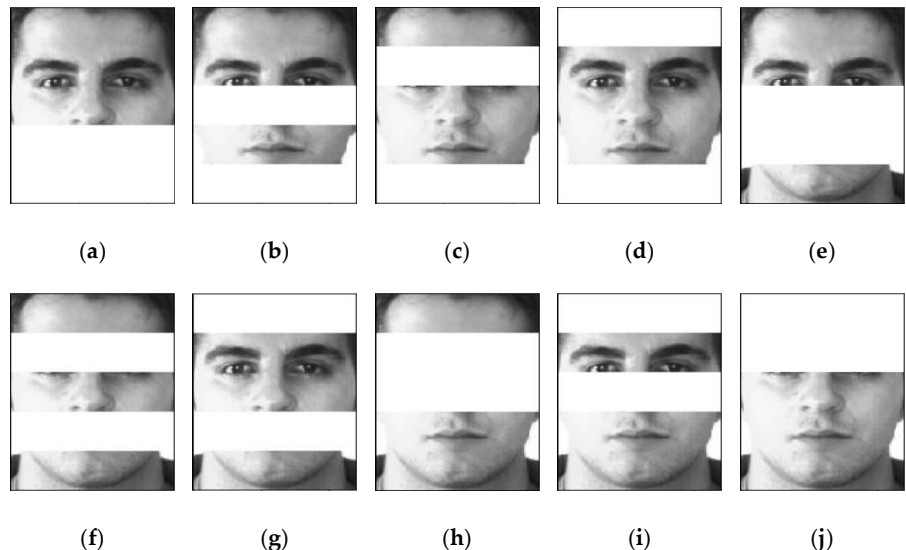

**Figure 3.** Permutation scheme for the third group of training sub-images: (**a**) reserved block 1, 2, 3; (**b**) reserved block 1, 2, 4; (**c**) reserved block 1, 3, 4; (**d**) reserved block 2, 3, 4; (**e**) reserved block 1, 2, 5; (**f**) reserved block 1, 3, 5; (**g**) reserved block 2, 3, 5; (**h**) reserved block 1, 4, 5; (**i**) reserved block 2, 4, 5; (**j**) reserved block 3, 4, 5 [Reprinted with permission from Elsevier [20]. Copyright 2013, Neurocomputing].

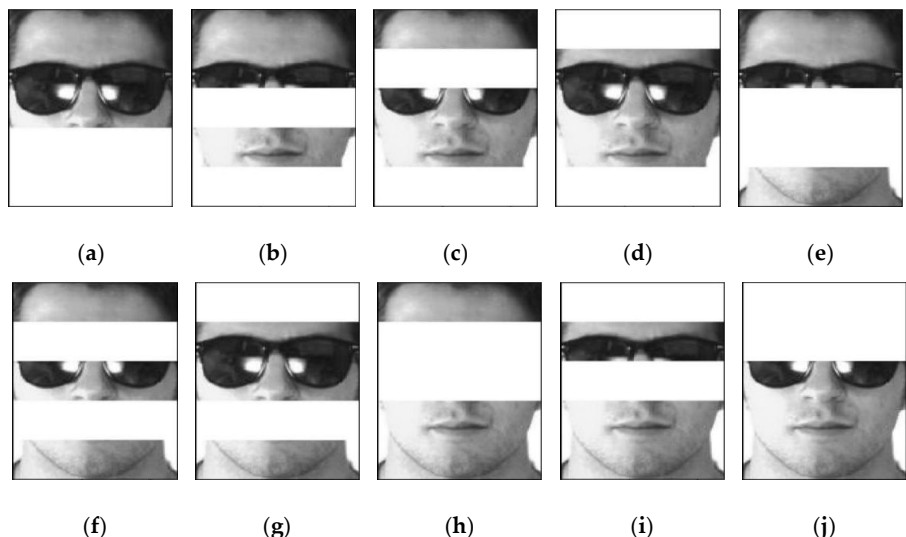

**Figure 4.** Arrangement and combination scheme of a certain test sub-image (involving sunglasses) in the third group: (**a**) reserved block 1, 2, 3; (**b**) reserved block 1, 2, 4; (**c**) reserved block 1, 3, 4; (**d**) reserved block 2, 3, 4; (**e**) reserved block 1, 2, 5; (**f**) reserved block 1, 3, 5; (**g**) reserved block 2, 3, 5; (**h**) reserved block 1, 4, 5; (**i**) reserved block 2, 4, 5; (**j**) reserved block 3, 4, 5 [Reprinted with permission from Elsevier [20]. Copyright 2013, Neurocomputing].

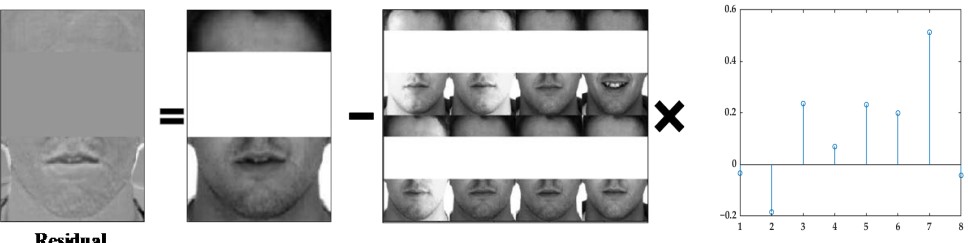

**Figure 5.** Residual calculation of arrangement scheme h (sunglasses occluding the face). [reprinted with permission from Elsevier [20]. Copyright 2013, Neurocomputing].

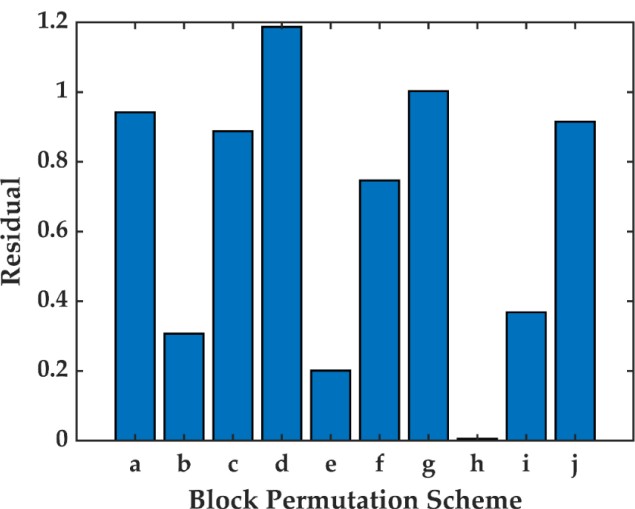

**Figure 6.** Residuals of face images involving sunglasses with different permutations [authors' own processing].

It can be seen from Figure 4 that this group retains 60% of the image information, and the sunglasses part was removed in scheme h. The distance between the test sample and the subspace projected by all training samples of an object was considered as the basis for the selection of the scheme. It can be seen from Figure 6 that the residual value of scheme h (preserved block positions one, four, and five) was the smallest. This means that the reorganized training samples in Figure 3h were constructed as an optimal linear model to predict the category of the test image in Figure 4h. Thus, the prediction result of scheme h was selected at this time.

The test sample contained face images involving scarves as an example, and the situation after the reorganization is shown in Figure 7. The residual calculation of the arrangement of scheme a is shown in Figure 8, and Figure 9 shows the minimum residuals of face images involving scarves with different permutations.

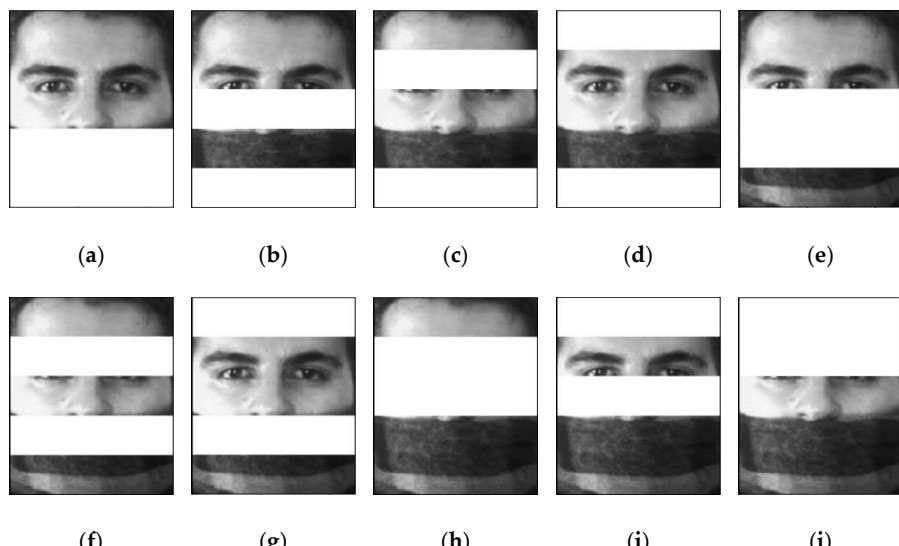

**Figure 7.** Arrangement and combination scheme of a certain test sub-image (involving a scarf) in the third group: (**a**) reserved block 1, 2, 3; (**b**) reserved block 1, 2, 4; (**c**) reserved block 1, 3, 4; (**d**) reserved block 2, 3, 4; (**e**) reserved block 1, 2, 5; (**f**) reserved block 1, 3, 5; (**g**) reserved block 2, 3, 5; (**h**) reserved block 1, 4, 5; (**i**) reserved block 2, 4, 5; (**j**) reserved block 3, 4, 5 [Reprinted with permission from Elsevier [20]. Copyright 2013, Neurocomputing].

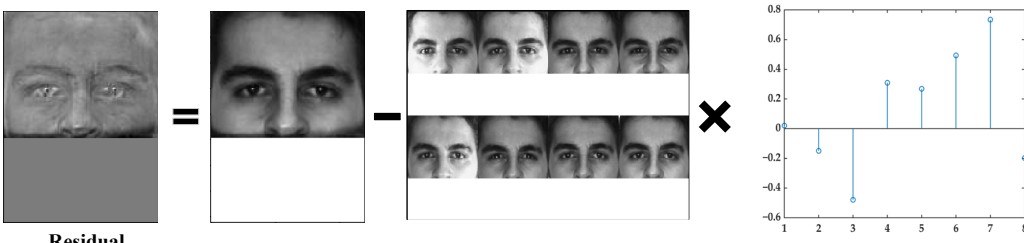

**Residual**

**Figure 8.** Residual calculation of arrangement scheme a (scarves occluding the face). [Reprinted with permission from Elsevier [20]. Copyright 2013, Neurocomputing].

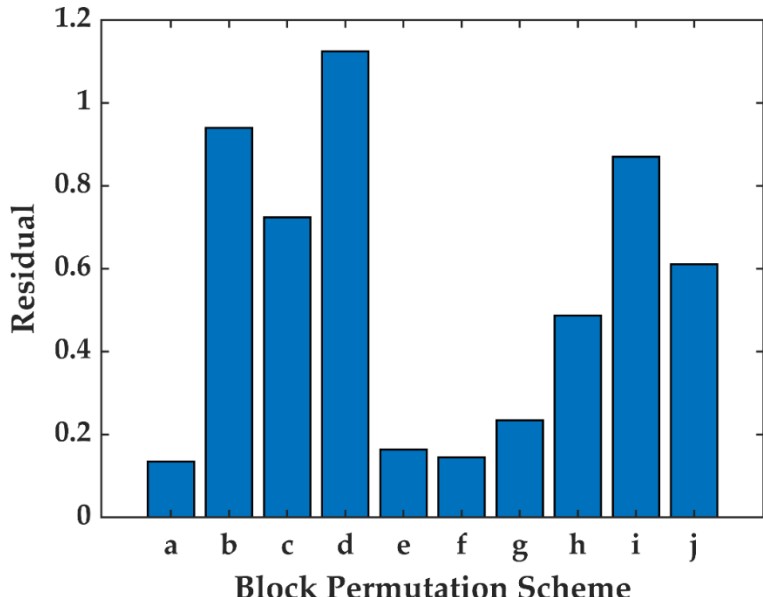

**Figure 9.** Residuals of face images involving scarves with different permutations [authors' own processing].

As shown in Figure 7, the scarf part was removed in scheme a. It can be seen from the residual values of each scheme in Figure 9 that the residual value of scheme a (reserved block positions one, two, and three) was the smallest. This means that the reconstituted training samples in Figure 3a were built using the optimal linear model to predict the class of the test image in Figure 7a. Therefore, the prediction result of scheme a was selected at this time. Similarly, the optimal arrangement and combination of one, two, four, and five sub-images were also similar to the above examples.

*3.3. Continuous Occlusion Result for the AR Database*

According to the above selection, the optimal block arrangement and combination was applied to the AR datasets (this method was also applied to the Extended Yale B and ORL datasets later). First of all, for data processing, the experiment divided the test analysis into three parts. The first part was the recognition of face images occluded by scarves. The second part was the recognition of face images occluded by sunglasses. The third part was the analysis of face images occluded by scarves and by sunglasses. When the BPLRC algorithm was applied, the face images were all horizontally divided into five blocks. The results regarding AR dataset recognition are shown in Table 1. In the table, ESRC stands for the Euler Sparse Representation Classification (ESRC) [17] algorithm. The experimental parameters were set according to [17], where $\lambda = 1.9$ and $\alpha = 0.5$. The results showed that the proposed BPLRC method was significantly better than the LRC, SRC, CRC, ESRC, and Module LRC algorithms in the three parts of the AR database. Among them, the LRC method was easily affected by the blocking features. In the AR datasets, the scarf part of the test image was linked to the characteristics of the male beard in the training

image. Learning the wrong characteristics caused model recognition performance to be poor. Although the SRC, ESRC, and CRC algorithms can also connect the scarf parts with the characteristics of a male beard, the three coefficients will be constrained through the L1 and L2 models, thus limiting their ability to learn in the wrong direction to a certain extent. Therefore, the accuracy rate of the SRC, ESRC, and CRC algorithms in terms of recognizing scarves covering human faces in images was higher than the LRC algorithm. The Module LRC and BPLRC methods extract the effective face characteristics as much as possible, and they will thus only learn a little noise information, as with the LRC, SRC, ESRC, and CRC algorithms. The BPLRC method is similar to Module LRC. The former considers retaining more blocks. However, face information was more effective than the Module LRC method, so the linear models learned more face characteristics. Therefore, the BPLRC algorithm identifies the effect of occlusion in face images better than other related algorithms.

**Table 1.** Accuracy (%) of different methods when identifying scarf occlusion, sunglasses occlusion, and mixed occlusion images.

| Occlusion Test | Image Size | Method | | | | | |
|---|---|---|---|---|---|---|---|
| | | LRC | SRC | CRC | ESRC | Module LRC | BPLRC |
| Scarf occlusion | $15 \times 10$ | 8.33 | 31.33 | 25.67 | 7.67 | 37.00 | **85.33** |
| | $20 \times 15$ | 10.33 | 42.67 | 46.67 | 18.33 | 60.67 | **91.33** |
| | $25 \times 20$ | 11.33 | 48.00 | 61.67 | 22.33 | 67.33 | **93.67** |
| Sunglasses occlusion | $15 \times 10$ | 37.00 | 25.00 | 20.33 | 24.00 | 71.67 | **85.67** |
| | $20 \times 15$ | 51.33 | 48.00 | 46.33 | 38.67 | 86.00 | **91.00** |
| | $25 \times 20$ | 54.33 | 47.33 | 47.00 | 39.67 | 88.00 | **90.67** |
| Mixed occlusion | $15 \times 10$ | 22.67 | 28.17 | 23.00 | 15.84 | 54.34 | **85.50** |
| | $20 \times 15$ | 30.83 | 45.34 | 46.50 | 28.50 | 73.34 | **91.17** |
| | $25 \times 20$ | 32.83 | 47.67 | 54.34 | 31.00 | 77.67 | **92.17** |

The LRC and CRC methods are relatively simple, and the calculation time is short. For $25 \times 20$ pixels images, the average time for each image with the LRC and CRC methods was 0.61 s and 0.98 s. The advantage of the LRC method is that it is very simple to calculate; however, it is easily affected by abnormal values. When the test graph contains continuous occlusion, recognition performance decreases sharply. The BPLRC method proposed in this article improves its robustness based on LRC, but the calculation cost increases. From Table 2, it was found that the calculation time of SRC and ESRC was much higher than the BPLRC method, and Table 1 did not show robustness higher than BPLRC.

**Table 2.** Computational time (seconds) for different methods to identify 300 images.

| Image Size | Method | | | | | |
|---|---|---|---|---|---|---|
| | LRC | SRC | CRC | ESRC | Module LRC | BPLRC |
| $15 \times 10$ | **0.46** | 55.46 | 0.78 | 61.03 | 2.36 | 15.83 |
| $20 \times 15$ | **0.46** | 143.15 | 0.78 | 140.65 | 3.49 | 21.74 |
| $25 \times 20$ | **0.61** | 323.80 | 0.98 | 329.51 | 4.92 | 28.17 |

*3.4. Robust Regression Model Based on Occlusion Training*

In practice, there may be too few images in each object, and these images contain a lot of non-face information. A face image with a scarf-occluded face was used as the training sample, and a face image without occlusion but with different lighting conditions was used as the recognition object (see Figure 10). Table 3 shows the results of identifying downsampled images with a resolution of $25 \times 20$ pixels. The results in Table 3 and Figure 11 prove that the algorithm has strong robustness and outperforms other algorithms, even with a small number of training samples.

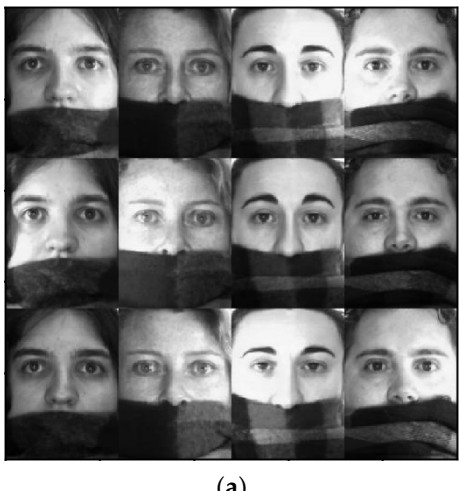 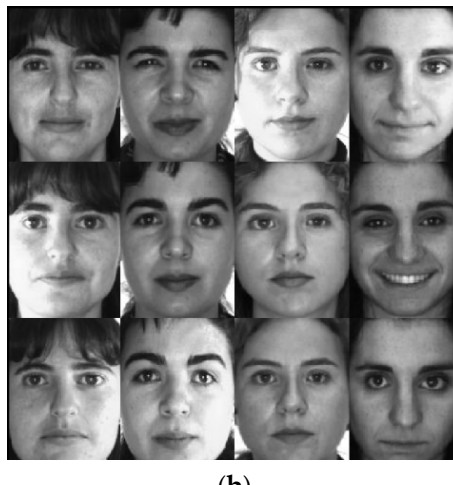

|             |             |
| :---------: | :---------: |
| (**a**)     | (**b**)     |

**Figure 10.** Some face images from the AR subset: (**a**) the left picture shows some of the training images used in the experiment; (**b**) the right picture shows some of the test samples used in the experiment [Reprinted with permission from Elsevier [20]. Copyright 2013, Neurocomputing].

**Table 3.** Accuracy of different methods when recognizing face images with scarves (images are downsampled to 500 dimensions).

| Method     | Number of Training Samples Per Class | | | | |
| :--------: | :---: | :---: | :---: | :---: | :---: |
|            | **2** | **3** | **4** | **5** | **6** |
| LRC        | 6.75  | 7.00  | 10.75 | 9.88  | **10.88** |
| SRC        | 3.88  | 2.38  | 3.88  | 10.88 | **26.00** |
| CRC        | 24.00 | 28.63 | 51.13 | 59.25 | **61.00** |
| Module LRC | 20.25 | 49.13 | 63.00 | 73.00 | **80.75** |
| BPLRC      | 32.75 | 59.88 | 74.75 | 83.88 | **89.25** |

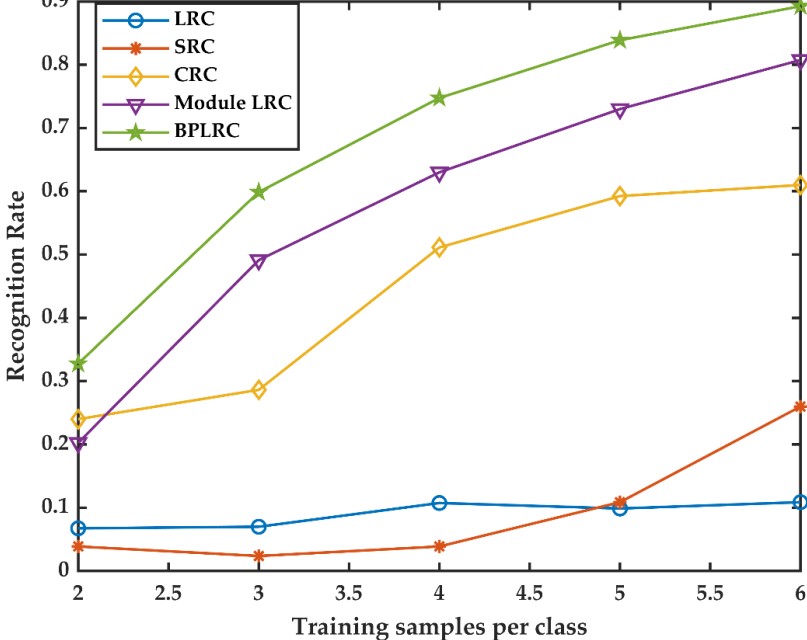

**Figure 11.** The relationship between the recognition rate of different methods for face images with scarves (the images were downsampled to 500 dimensions) and the number of training samples per class [authors' own processing].

### 3.5. Analysis of Verification Results of Different Data

3.5.1. Continuous Occlusion Results for the Extended Yale B Database

The Extended Yale B [23] face database is comprised of 64 face images per object that contain different lighting conditions (see Figure 12). There are a total of 38 objects, of which the 11th and 13th objects have only 60 images. The 12th contains 59, the 15th has 62, and the 14th, 16th, and 17th have 63. Therefore, in the experiment, the first 26 images of each object were selected as training samples, and the last 33 images were randomly occluded as test samples. A total of 38 objects were selected. Each image was downsampled from the original $165 \times 120$ pixels to $15 \times 10$ pixels, $20 \times 15$ pixels, and $25 \times 20$ pixels for the analysis.

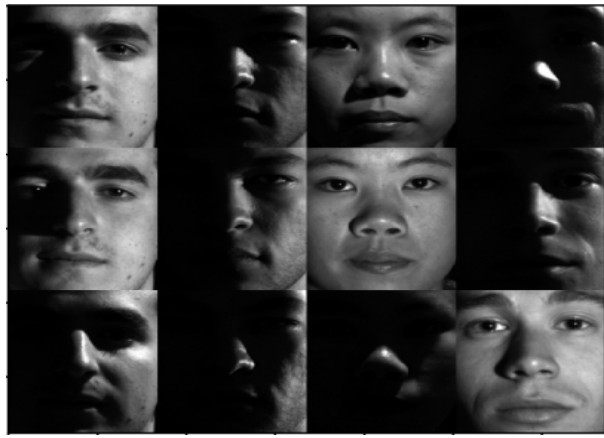

**Figure 12.** Some face images from Extended Yale B [Reprinted with permission from Elsevier [20]. Copyright 2013, Neurocomputing].

Figure 13 shows occlusion maps of different proportions that were used with the test samples. The results of Extended Yale B database recognition are shown in Tables 4 and 5. The results in the table show that the proposed BPLRC method was significantly better than the LRC, SRC, CRC, ESRC, and Module LRC (five-block processing) algorithms. For $25 \times 20$ pixels images, the average time taken for the BPLRC algorithm to identify each image was 0.08 s (see Table 6), and the calculation time was not very long. However, lengthening the calculation time of the LRC algorithm is sometimes necessary in exchange for higher robustness. For example, in Table 4, the higher the occlusion rate of the test image, the greater the difference between the accuracy of the BPLRC and LRC methods.

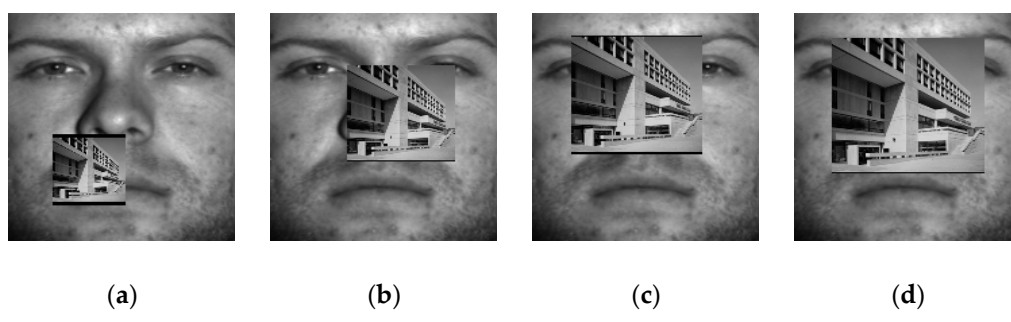

|         (a)         |         (b)         |         (c)         |         (d)         |

**Figure 13.** Occlusion maps of different proportions used with the test samples: (**a**) face image with an occlusion ratio of 10%; (**b**) face image with an occlusion ratio of 20%; (**c**) face image with an occlusion ratio of 30%; (**d**) face image with an occlusion ratio of 40% [Reprinted with permission from Elsevier [20]. Copyright 2013, Neurocomputing].

**Table 4.** Accuracy (%) of different methods when identifying 10%-, 20%-, 30%-, and 40%-occluded face images in the Extended Yale B database.

| Occlusion Rate | Image Size | Method | | | | | |
|---|---|---|---|---|---|---|---|
| | | LRC | SRC | CRC | ESRC | Module LRC | BPLRC |
| 10% | $15 \times 10$ | 60.21 | 47.29 | 43.06 | 33.25 | 4.94 | **64.51** |
| | $20 \times 15$ | 65.15 | 64.83 | 60.05 | 47.45 | 32.46 | **77.27** |
| | $25 \times 20$ | 66.83 | 67.78 | 67.46 | 46.73 | 68.42 | **78.87** |
| 20% | $15 \times 10$ | 43.46 | 41.63 | 35.89 | 28.79 | 5.02 | **59.57** |
| | $20 \times 15$ | 52.47 | 55.74 | 52.23 | 41.87 | 32.30 | **74.40** |
| | $25 \times 20$ | 57.58 | 59.49 | 61.48 | 40.91 | 67.62 | **76.00** |
| 30% | $15 \times 10$ | 26.95 | 31.58 | 27.51 | 23.52 | 5.26 | **51.28** |
| | $20 \times 15$ | 40.83 | 45.69 | 40.99 | 33.73 | 30.78 | **71.85** |
| | $25 \times 20$ | 46.33 | 45.61 | 50.80 | 34.53 | 68.66 | **74.00** |
| 40% | $15 \times 10$ | 21.29 | 25.36 | 22.57 | 19.70 | 4.70 | **40.19** |
| | $20 \times 15$ | 29.11 | 32.70 | 32.69 | 27.43 | 30.46 | **66.59** |
| | $25 \times 20$ | 37.16 | 33.25 | 39.95 | 26.95 | 67.30 | **68.74** |

**Table 5.** Accuracy (%) of different methods when identifying 10% vertically and 20% diagonally occluded face images in the Extended Yale B database.

| Occlusion Rate and Method | Image Size | Method | | | | | |
|---|---|---|---|---|---|---|---|
| | | LRC | SRC | CRC | ESRC | Module LRC | BPLRC |
| 10% (Vertical Occlusion) | $15 \times 10$ | 58.21 | 51.51 | 46.81 | 31.34 | 3.11 | **64.19** |
| | $20 \times 15$ | 65.15 | 66.91 | 63.08 | 45.21 | 54.70 | **70.41** |
| | $25 \times 20$ | 67.62 | 70.41 | 71.05 | 46.73 | 56.86 | **74.72** |
| 20% (Diagonal Occlusion) | $15 \times 10$ | 33.25 | **40.27** | 33.01 | 24.80 | 3.35 | 33.25 |
| | $20 \times 15$ | 40.43 | 51.12 | 54.39 | 39.39 | 42.50 | **75.44** |
| | $25 \times 20$ | 46.25 | 63.32 | 63.32 | 35.33 | 69.06 | **76.79** |

**Table 6.** Computational time (seconds) for different methods to identify 1254 images.

| Image Size | Method | | | | | |
|---|---|---|---|---|---|---|
| | LRC | SRC | CRC | ESRC | Module LRC | BPLRC |
| $15 \times 10$ | **0.83** | 248.20 | 1.40 | 255.64 | 6.89 | 45.69 |
| $20 \times 15$ | **0.9** | 770.95 | 1.59 | 860.85 | 12.67 | 72.87 |
| $25 \times 20$ | **1.04** | 2231.74 | 2.03 | 2073.83 | 15.08 | 100.86 |

However, randomly generating occlusion blocks to occlude the faces in the Extended Yale B dataset needs to be more comprehensive. The two special cases of vertical and diagonal occlusion also need to be considered. The vertical occlusion of 10% of faces is shown in Figure 14a,b, and the diagonal occlusion of 20% of faces is shown in Figure 14c,d.

It can be seen from Table 5 that, compared to the recognition accuracy of BPLRC when images are randomly occluded by 10%, in most cases, the recognition accuracy of BPLRC is higher than that of LRC, SRC, CRC, ESRC, and Module LRC. For example, in the case of 20% diagonal occlusion, the SRC method has better recognition accuracy than BPLRC for images downsampled to $15 \times 10$ pixels. The biggest reason is the particularity of the occlusion distribution and the small size of the image after downsampling. The results in Table 5 show that, with the diagonal face occlusion in this special case, good results can also be obtained when using the BPLRC method.

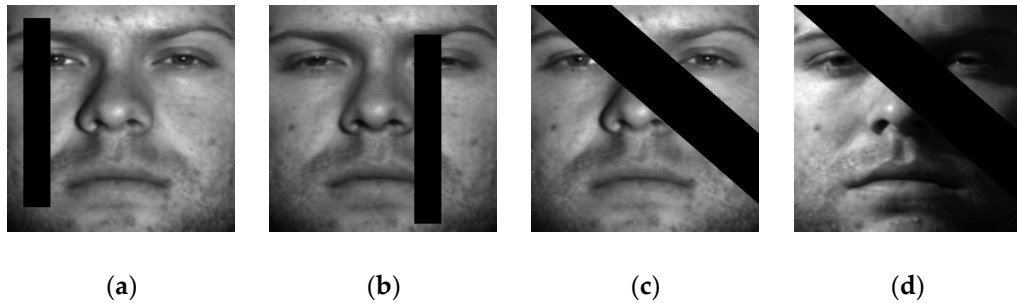

|   (**a**)   |   (**b**)   |   (**c**)   |   (**d**)   |

**Figure 14.** Test samples with vertical occlusion of 10% and diagonal occlusion of 20%: (**a**,**b**) face images with 10% vertical occlusion; (**c**,**d**) face images with 20% diagonal occlusion [Reprinted with permission from Elsevier [20]. Copyright 2013, Neurocomputing].

3.5.2. Continuous Occlusion Results with the ORL Database

There were 10 grayscale images per object in the ORL [24] face database for a total of 40 objects. Some of these images were different in terms of shooting time, lighting, facial expressions (eyes open/closed, smiling), and facial details (glasses). All images in the ORL database were selected for the experiment. The first six images were used as training samples. The last four images were randomly occluded as test samples (see Figure 15). The pixel size of each image was downsampled from the original $112 \times 92$ pixels to $15 \times 10$ pixels, $20 \times 15$ pixels, and $25 \times 20$ pixels for the experiments.

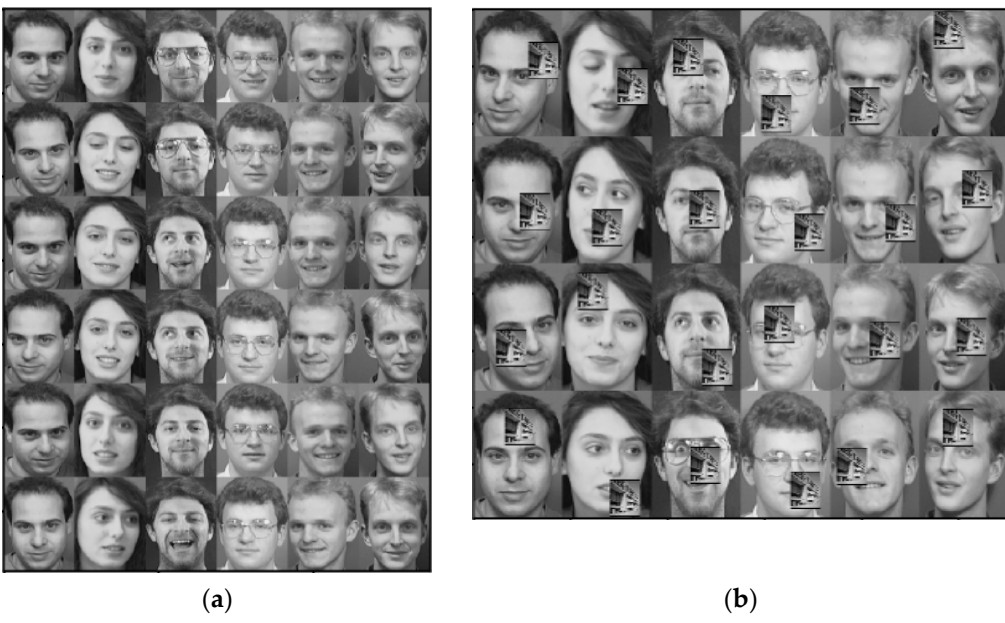

|   (**a**)   |   (**b**)   |

**Figure 15.** Some images from the ORL database: (**a**) the ORL database without occluded images; (**b**) the ORL database with 10%-occluded images [Reprinted with permission from Elsevier [20]. Copyright 2013, Neurocomputing].

The processing method for the ORL database was consistent with the Extended Yale B database. The results of ORL database recognition are shown in Tables 7 and 8. Table 9 shows the calculation time for different methods to identify 160 images in the ORL dataset. The proposed BPLRC method also significantly outperformed the LRC, SRC, CRC, ESRC, and Module LRC (five-block processing) algorithms when using this database. The computation time of the BPLRC method is moderate, and its computational complexity is lower than that of the SRC and ESRC methods. Synthesizing the results of BPLRC on the AR, Extended Yale B, and ORL datasets demonstrated the effectiveness of the proposed BPLRC for solving face occlusion problem.

**Table 7.** Accuracy (%) of different methods when identifying 10%-, 20%-, 30%-, and 40%-occluded face images from the ORL database.

| Occlusion Rate | Image Size | Method | | | | | |
| --- | --- | --- | --- | --- | --- | --- | --- |
| | | LRC | SRC | CRC | ESRC | Module LRC | BPLRC |
| 10% | 15 × 10 | 83.13 | 63.13 | 73.13 | 78.75 | 39.38 | **93.75** |
| | 20 × 15 | 90.63 | 30.00 | 75.63 | 35.00 | 76.25 | **96.88** |
| | 25 × 20 | 91.25 | 69.38 | 83.13 | 75.63 | 82.50 | **95.63** |
| 20% | 15 × 10 | 78.13 | 46.25 | 61.25 | 59.38 | 41.25 | **88.75** |
| | 20 × 15 | 82.50 | 15.63 | 68.75 | 23.75 | 78.13 | **93.13** |
| | 25 × 20 | 88.75 | 53.75 | 71.88 | 54.38 | 84.38 | **93.75** |
| 30% | 15 × 10 | 56.25 | 36.25 | 50.00 | 51.51 | 37.50 | **85.63** |
| | 20 × 15 | 65.00 | 13.75 | 52.50 | 13.75 | 76.25 | **93.13** |
| | 25 × 20 | 69.38 | 43.13 | 58.13 | 43.75 | 80.00 | **91.88** |
| 40% | 15 × 10 | 39.38 | 23.75 | 35.00 | 34.38 | 34.38 | **83.75** |
| | 20 × 15 | 45.00 | 10.00 | 40.00 | 11.88 | 75.63 | **88.13** |
| | 25 × 20 | 46.25 | 29.38 | 48.13 | 29.38 | 83.75 | **88.75** |

**Table 8.** Accuracy (%) of different methods when identifying 10% vertically and 20% diagonally occluded face images from the ORL database.

| Occlusion Rate and Method | Image Size | Method | | | | | |
| --- | --- | --- | --- | --- | --- | --- | --- |
| | | LRC | SRC | CRC | ESRC | Module LRC | BPLRC |
| 10% (Vertical Occlusion) | 15 × 10 | 89.38 | 36.88 | 53.13 | 58.75 | 85.63 | **91.25** |
| | 20 × 15 | 85.63 | 17.50 | 60.00 | 24.38 | 91.25 | **95.00** |
| | 25 × 20 | 89.38 | 47.50 | 63.13 | 55.00 | 90.63 | **93.13** |
| 20% (Diagonal Occlusion) | 15 × 10 | 23.75 | 13.75 | 30.63 | 31.25 | 53.13 | **76.25** |
| | 20 × 15 | 28.13 | 5.63 | 36.88 | 3.75 | 85.63 | **85.63** |
| | 25 × 20 | 27.50 | 17.50 | 26.88 | 12.50 | 85.63 | **85.63** |

**Table 9.** Computational time (seconds) for different methods to identify 160 images.

| Image Size | Method | | | | | |
| --- | --- | --- | --- | --- | --- | --- |
| | LRC | SRC | CRC | ESRC | Module LRC | BPLRC |
| 15 × 10 | **0.16** | 9.30 | 0.19 | 11.16 | 0.48 | 2.97 |
| 20 × 15 | **0.10** | 40.83 | 0.16 | 22.23 | 0.60 | 3.46 |
| 25 × 20 | **0.13** | 24.08 | 0.22 | 20.70 | 0.66 | 4.06 |

In the ORL datasets, the two special cases of vertical occlusion of the face and diagonal occlusion of the face were also considered. Vertical occlusion of 10% of the face is shown in Figure 16a,b, and diagonal occlusion of 20% of the face is shown in Figure 16c,d.

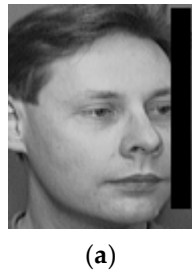 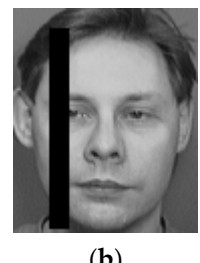 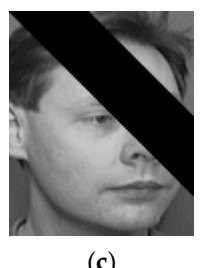 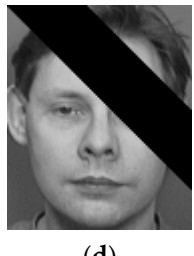

(**a**)　　　　　　　(**b**)　　　　　　　(**c**)　　　　　　　(**d**)

**Figure 16.** Test samples with vertical occlusion of 10% and diagonal occlusion of 20% from the ORL datasets: (**a**,**b**) face images with 10% vertical occlusion; (**c**,**d**) face images with 20% diagonal occlusion [Reprinted with permission from Elsevier [20]. Copyright 2013, Neurocomputing].

It can be seen from Table 8 that, compared to face images randomly obscured by 10% and 20%, the recognition accuracy of the BPLRC method has decreased. However, the BPLRC method also obtains the best results when compared to other related methods.

## 4. Discussion

As has been reported in the literature [10,11,14,18], LRC, SRC, and CRC use the residues between the Euclidean distance measurement model and the test image. There are advantages and disadvantages to each. From the above experimental results, ESRC identification is, in many cases, inferior to the SRC method. The proposed BPLRC algorithm aims to determine the characteristics of the best face recognition and then use the LRC method to classify them. Therefore, the measurement method of the BPLRC algorithm is Euclidean distance.

To fully verify that the BPLRC method's recognition performance is better than other methods, in addition to identifying accuracy indicators this chapter introduces precision, recall, and F1score indicators to evaluate the model. Because AR, Extended Yale B, and ORL datasets include many categories, each category cannot be evaluated locally. All the methods involved in this paper were evaluated globally by calculating precision, recall, and F1score, which were >0.7 in all categories. When precision, recall, and F1score are equal to 0, the samples representing a certain category are incorrectly identified. The more of such categories there are, the worse the model's performance.

### 4.1. Continuous Occlusion Analysis of the AR Database

The AR dataset contains 100 classes of target faces. Discussing the assessment metrics for all categories takes time and effort. We set the threshold for precision, recall, and F1score to 0.7 and then analyzed the number of categories that were greater than 0.7. If the threshold was too large or too small, the value was close and it was not easy to compare the performance of the following methods.

As shown in Table 10, BPLRC identifies face images containing only scarf occlusion, sunglasses occlusion, or mixed occlusion whose number of precision, recall, and F1score > 0.7 categories is more than or equal to other methods. Precision, recall, and F1score equal to 0 was present in less categories than other methods. The model evaluation indicators involved in the table, combined with the identification accuracy of the experimental part, show that the proposed method can effectively solve the face occlusion problem in the AR face dataset. However, other relevant algorithms, such as the LRC, SRC, CRC, and ESRC methods, have poor identification performance, partly because they belong to linear models and are, in turn, susceptible to anomalous variables or abnormal points. The key to the SRC, CRC, and ESRC methods, compared to the LRC method, is that they regularize the model coefficient, which improves the robustness of the linear model to some extent. From the results relating to identification of scarf occlusion, the SRC, CRC, and ESRC methods are much better than the LRC method. The results relating to identification of sunglasses occlusion show that the LRC method is superior to the SRC, CRC, and ESRC methods. This indicates the effect of a face image linearly represented by the same category of face image, which is generally better than a face image linearly represented by all categories of face images. To circumvent this drawback, that LRC is extremely poor in robustness, block arrangement is combined with the LRC method. Block arrangement achieves excellent recognition performance in LRC and ensures that more useful face information is extracted. Module LRC only retains the most useful block, thus keeping too little effective information which further leads to its inferior recognition performance compared to BPLRC.

**Table 10.** The number of categories that identified performance indicators to an extent greater than 0.7 or single categories that were wrongly identified from AR datasets for different methods.

| Occlusion Test | Other Measurement Model Performance Indicators | Method | | | | | |
|---|---|---|---|---|---|---|---|
| | | LRC | SRC | CRC | ESRC | Module LRC | BPLRC |
| Scarf occlusion | Precision > 0.7 | 6 | 24 | 38 | 4 | 77 | **86** |
| | Recall > 0.7 | 8 | 46 | 70 | 24 | 91 | **94** |
| | F1Score > 0.7 | 5 | 33 | 53 | 6 | 90 | **93** |
| | Precision, recall, and F1Score = 0 | 81 | 24 | 15 | 50 | 2 | **1** |
| Sunglasses occlusion | Precision > 0.7 | 37 | 27 | 25 | 13 | 72 | **75** |
| | Recall > 0.7 | 54 | 48 | 54 | 41 | 85 | **93** |
| | F1Score > 0.7 | 42 | 30 | 43 | 23 | 89 | **94** |
| | Precision, recall, and F1Score = 0 | 29 | 29 | 15 | 31 | 2 | **0** |
| Mixed occlusion | Precision > 0.7 | 4 | 15 | 34 | 1 | 93 | **96** |
| | Recall > 0.7 | 5 | 51 | 26 | 26 | 86 | **91** |
| | F1Score > 0.7 | 45 | 20 | 69 | 1 | 93 | **93** |
| | Precision, recall, and F1Score = 0 | 23 | 7 | 7 | 16 | 1 | **0** |

*4.2. Extended Yale B Database Analysis and Discussion*

Extended Yale B datasets include 38 target faces. As shown in Table 11, when BPLRC recognizes 10%, 20%, and 30% of face images, its number of categories where precision, recall, and F1score are >0.7 is more than other methods. When BPLRC identifies 40% of the face image, its number of precision categories >0.7 is more than other methods, but the number of recall and F1score categories > 0.7 is less than the Module LRC method. In fact, in face recognition, precision indicators are more important than recall and F1Score. For example, when the precision value is too low in the family access control system, it is possible to identify strangers as family members. When the recall value is too low, even family members cannot be identified (though you can enter the house by inputting a password), and there will be no severe theft incidents. F1score is the harmonic mean of two metrics, and simply provides a summary evaluation of model performance. According to the results in Table 11, the number of precision, recall, and F1score categories equal to 0 is only 1 or 0; thus, the model's identification performance cannot be evaluated from the number of precision, recall, or F1Score categories equal to 0.

**Table 11.** The number of categories that identified performance indicators to an extent greater than 0.7 or single categories that were wrongly identified from Extended Yale B datasets for different methods.

| Occlusion Rate and Method | Other Measurement Model Performance Indicators | Method | | | | | |
|---|---|---|---|---|---|---|---|
| | | LRC | SRC | CRC | ESRC | Module LRC | BPLRC |
| 10% (Random Occlusion) | Precision > 0.7 | 16 | 12 | 16 | 0 | 17 | **30** |
| | Recall > 0.7 | 28 | 19 | 31 | 4 | 24 | **31** |
| | F1Score > 0.7 | 17 | 15 | 22 | 0 | 16 | **32** |
| | Precision, recall, and F1Score = 0 | 0 | 0 | 0 | 0 | 0 | **0** |
| 20% (Random Occlusion) | Precision > 0.7 | 8 | 10 | 12 | 0 | 15 | **29** |
| | Recall > 0.7 | 21 | 13 | 24 | 5 | 22 | **28** |
| | F1Score > 0.7 | 7 | 7 | 12 | 0 | 16 | **31** |
| | Precision, recall, and F1Score = 0 | 0 | 0 | 0 | 0 | 0 | **0** |
| 30% (Random Occlusion) | Precision > 0.7 | 4 | 4 | 4 | 0 | 16 | **25** |
| | Recall > 0.7 | 21 | 2 | 19 | 2 | 24 | **25** |
| | F1Score > 0.7 | 2 | 1 | 4 | 0 | 15 | **27** |
| | Precision, recall, and F1Score = 0 | 0 | 0 | 0 | 0 | 0 | **0** |
| 40% (Random Occlusion) | Precision > 0.7 | 2 | 0 | 2 | 0 | 10 | **17** |
| | Recall > 0.7 | 12 | 1 | 12 | 1 | **29** | 24 |
| | F1Score > 0.7 | 0 | 0 | 1 | 0 | **25** | 17 |
| | Precision, recall, and F1Score = 0 | 0 | 0 | 0 | 0 | 0 | **0** |

**Table 11.** *Cont.*

| Occlusion Rate and Method | Other Measurement Model Performance Indicators | Method | | | | | |
|---|---|---|---|---|---|---|---|
| | | LRC | SRC | CRC | ESRC | Module LRC | BPLRC |
| 10% (Vertical Occlusion) | Precision > 0.7 | 18 | 20 | 18 | 2 | 8 | **25** |
| | Recall > 0.7 | 30 | 20 | **30** | 7 | 25 | 29 |
| | F1Score > 0.7 | 19 | 22 | 24 | 0 | 4 | **27** |
| | Precision, recall, and F1Score = 0 | 0 | 0 | 0 | 0 | 0 | **0** |
| 20% (Diagonal Occlusion) | Precision > 0.7 | 7 | 17 | 14 | 2 | 18 | **26** |
| | Recall > 0.7 | 17 | 19 | **28** | 8 | 25 | 27 |
| | F1Score > 0.7 | 4 | 13 | 17 | 1 | 17 | **28** |
| | Precision, recall, and F1Score = 0 | 1 | 0 | 0 | 0 | 0 | **0** |

In summary, from the number of categories for each method with precision >0.7, the proposed method identifies 10%, 20%, 30%, and 40% of faces, vertical occlusion of 10% of faces, and diagonal occlusion of 20% of face images better than other related algorithms.

### 4.3. ORL Database Analysis and Discussion

The ORL dataset contains 40 classes of target faces. As shown in Table 12, when BPLRC identifies random occlusion of 10%, 20%, and 30% of faces, 10% vertical occlusion of faces, and 20% diagonal occlusion of faces, its number of precision, recall, and F1score categories greater than 0.7 is greater than other related methods. The number of precision, recall, and F1score categories equal to 0 is less than or equal to different related algorithms. Therefore, the BPLRC method solves the face occlusion problem more effectively in the ORL datasets than the other related algorithms, showing stronger robustness than LRC, SRC, CRC, ESRC, and Module LRC. Combined with the identification accuracy obtained from the previous experiments, we fully confirm that BPLRC improves upon both LRC and Module LRC.

**Table 12.** The number of categories that identified performance indicators to an extent greater than 0.7 or single categories that were wrongly identified from ORL datasets for different methods.

| Occlusion Rate and Method | Other Measurement Model Performance Indicators | Method | | | | | |
|---|---|---|---|---|---|---|---|
| | | LRC | SRC | CRC | ESRC | Module LRC | BPLRC |
| 10% (Random Occlusion) | Precision > 0.7 | 39 | 24 | 33 | 29 | 37 | **40** |
| | Recall > 0.7 | 38 | 27 | 32 | 27 | 37 | **40** |
| | F1Score > 0.7 | 39 | 18 | 30 | 25 | 37 | **40** |
| | Precision, recall, and F1Score = 0 | 0 | 1 | 0 | 1 | 0 | **0** |
| 20% (Random Occlusion) | Precision > 0.7 | 38 | 14 | 27 | 13 | 36 | **40** |
| | Recall > 0.7 | 38 | 13 | 26 | 22 | 37 | **39** |
| | F1Score > 0.7 | 37 | 9 | 23 | 10 | 36 | **40** |
| | Precision, recall, and F1Score = 0 | 0 | 4 | 1 | 2 | 0 | **0** |
| 30% (Random Occlusion) | Precision > 0.7 | 24 | 11 | 15 | 12 | 36 | **38** |
| | Recall > 0.7 | 26 | 10 | 17 | 15 | 34 | **38** |
| | F1Score > 0.7 | 16 | 5 | 8 | 7 | 34 | **38** |
| | Precision, recall, and F1Score = 0 | 1 | 7 | 1 | 6 | 0 | **0** |
| 40% (Random Occlusion) | Precision > 0.7 | 14 | 5 | 18 | 5 | 39 | **39** |
| | Recall > 0.7 | 18 | 6 | 18 | 11 | 37 | 37 |
| | F1Score > 0.7 | 9 | 1 | 10 | 2 | 37 | 37 |
| | Precision, recall, and F1Score = 0 | 7 | 13 | 6 | 12 | 0 | **0** |
| 10% (Vertical Occlusion) | Precision > 0.7 | 36 | 12 | 19 | 15 | 38 | **40** |
| | Recall > 0.7 | 36 | 15 | 24 | 25 | 36 | **39** |
| | F1Score> 0.7 | 36 | 8 | 15 | 11 | 36 | **39** |
| | Precision, recall, and F1Score = 0 | 0 | 3 | 0 | 1 | 0 | **0** |
| 20% (Diagonal Occlusion) | Precision > 0.7 | 9 | 5 | 8 | 2 | 35 | 35 |
| | Recall > 0.7 | 9 | 6 | 14 | 8 | 34 | 34 |
| | F1Score > 0.7 | 4 | 1 | 4 | 1 | 33 | 33 |
| | Precision, recall, and F1Score = 0 | 25 | 27 | 21 | 27 | 0 | **0** |

### 4.4. Comparison of Differences in Algorithms

The LRC method uses individual categories of training samples to represent the test samples linearly. In contrast, the SRC, CRC, and ESRC methods use all categories of training samples to represent the test samples linearly. Other differences are shown in Table 13, with SRC being equivalent to the Lasso regression model and constraining the regression coefficients using the L1 norm. CRC is equivalent to the ridge regression model, and the regression coefficient is constrained using the L2 norm. At the same time, ESRC replaces the metric of the SRC method with the Euler distance. The Module LRC and BPLRC methods both determine the face features conducive to the LRC method and further determine the category to which the image belongs. The BPLRC approach is similar to Module LRC in that it considers the case of block combinations.

**Table 13.** Attribute comparison with prior methods.

| Method | LRC [18] | SRC [10,11] | CRC [14] | ESRC [17] | Module LRC [18] | BPLRC |
|---|---|---|---|---|---|---|
| The basic model | Linear regression model | Lasso regression model | Ridge regression model | Lasso regression model | Linear regression model | Linear regression model |
| Measuring method | Euclidean distance | Euclidean distance | Euclidean distance | Euler distance | Euclidean distance | Euclidean distance |
| Image recognition speed | Extremely fast | Extremely slow | Fast | Extremely slow | Relatively fast | Relatively slow |
| Robustness | Extremely weak | Relatively weak | Relatively weak | Weak | Relatively strong | Strong |
| Scope | 1. Face image with light changes; 2. face image with expression changes | 1. Face image with light changes; 2. face image with expression changes | 1. Face image with light changes; 2. face image with expression changes | 1. Face image with light changes; 2. face image with expression changes | 1. Face image with light changes; 2. face image with expression changes; 3. face occlusion image | 1. Face image with light changes; 2. face image with expression changes; 3. face occlusion image |

Table 13 shows the differences in the LRC, SRC, CRC, ESRC, Module LRC, and BPLRC methods. The linear model is susceptible to contaminated data [26] and LRC is directly affected by contaminated data, while other RBCM algorithms are more robust than LRC algorithms. Notably, Module LRC and BPLRC can identify the target images by effectively using face features. If a small number of non-face images exist in the test sample, the influence function in the literature [27] is used to obtain a clean sample set. From the recognition results of the three face datasets, Module LRC and BPLRC are more suitable for recognizing facial occlusion images and have strong robustness. The three datasets contain various face images with light changes and expression changes. Therefore, all RBCM methods (including LRC, SRC, CRC, ESRC, Module LRC, and BPLRC) can achieve better results when setting a low occlusion ratio. At the same time, it shows that RBCM methods can effectively solve the problem of illumination and facial expression changes.

## 5. Conclusions

The method proposed in this paper combines local image information into a whole, reflecting both the local information and the overall information of the image. In the AR datasets, face images with scarves were downsampled to $25 \times 20$ pixels as an example, and the recognition accuracy of the BPLRC algorithm in identifying the face images with scarf occlusion was 93.67%. The number of categories with precision, recall, and F1score greater than 0.7 was 86, 94, and 93, respectively, and the number of categories with precision, recall,

and F1score equal to 0 was 1. These indicators can indicate the degree of excellence various algorithms have in identification. Furthermore, experiments on the Extended Yale B, ORL, and AR datasets showed that the BPLRC algorithm was significantly better than other related classification methods in identifying images with continuous occlusion. The LRC, SRC, ESRC, and CRC algorithms did not remove occlusions, which caused them to learn a lot of noise. As a result, model performance was worse than the Module LRC and BPLRC models. Although Module LRC removes the continuous occlusion part, it only considers retaining one block, and there may be less reserved effective face information than the BPLRC method. Therefore, the BPLRC algorithm's ability to identify face images is better than other related algorithms.

BPLRC reorganizes different training samples and test samples through block arrangement and combination, but the number of combinations increases exponentially with the number of blocks. Compared to the LRC algorithm, the method proposed optimizes face characteristics in the image while at the same time reducing the negative impact of occlusion on the model. For example, the LRC method easily attributes face images with scarf occlusion to a large number of beards, or other categories that have characteristics similar to scarves. Compared to the Module LRC algorithm, this algorithm's novelty lies in retaining as many image block schemes as possible in order to retain useful face characteristics. Additionally, the block arrangement can be combined with other algorithms that are less robust. For example, when the number of blocks is five, the average time taken with BPLRC in AR datasets for setting an image with a size of $25 \times 20$ pixels is 0.094 s, and the calculation amount is relatively small. However, when the number of image divisions is too large, many arrangement schemes will greatly reduce the recognition speed of the BPLRC algorithm. In response to the defects of the BPLRC algorithm, in the future, the rapid iteration method will be studied and the optimal or subsequent arrangement will be found in many solutions to reduce the calculation amount, which should provide the possibility of dividing more blocks.

**Author Contributions:** Conceptualization, data curation, software, validation, writing—original draft preparation, visualization, J.X.; conceptualization, writing—review and editing, funding acquisition, project administration and formal analysis, X.C. (Xiaojing Chen); conceptualization, writing—review and editing, and resources, Z.X.; writing—review and editing, and resources, S.A.; resources, L.Y. and X.C. (Xi Chen); writing—review and editing, formal analysis, and validation, W.S.; software, writing—review and editing, and supervision, G.H. All authors have read and agreed to the published version of the manuscript.

**Funding:** This research was funded by Wenzhou Social Development (Medical and Health) Science and Technology Project grant number [ZY2021027] and National Natural Science Foundation of China grant number [62105245, 61893096014].

**Institutional Review Board Statement:** Not applicable.

**Informed Consent Statement:** Informed consent was obtained from all subjects involved in the study.

**Data Availability Statement:** The data used to support the findings of this study are available from the corresponding author upon request.

**Acknowledgments:** The authors would like to acknowledge the financial support provided by the Natural Science Foundation of Zhejiang (LY21C200001 and LQ20F030059), the Wenzhou Science and Technology Bureau General Project (S2020011 and G20200044).

**Conflicts of Interest:** The authors declare that they have no conflict of interest.

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
