# Peer review of "Recognition of Continuous Face Occlusion Based on Block Permutation by Using Linear Regression Classification"

_applsci, doi:10.3390/app122311885_

Round 1

Reviewer 1 Report

1. Abstracts needs to be more concise about the importance of the proposed work and how it is achieved.

2. Introduction should discuss the importance of face occlusion. Various recent techniques and its disadvantages should be listed out.

3. Highlight the importance and need of the proposed methodology and how it is beneficial from existing methods.

4. Major contributions of the research should be signified with supportive measurements.

5. Discussion on related works is very poor. No recent approaches are discussed and analyzed for issues.

6. Schematic sketch of proposed work could be provided for improving the understanding from the reader side.

7. All the Equations are to be explained in detail with proper reference.

8. Equation 9 and 10 implies the block permutation analysis in face occlusion detection. Signify its importance.

9. Why euclidean distance calculation chosen for predicting the accuracy of the proposed method. Justify your choice of selection.

10. Grouping of sub spaces in images is performed on basis of which parameter? Why sub spacing is needed in your work?

11. What does graph in figure 2 signifies? No proper explanation available.

12. Why only horizontal occlusion is considered in face images? Why not vertical and diagonal occlusion?

13. What does figure 6 infers? No clear understanding of the image is provided.

14. Rather than accuracy, other performance should be considered for proving the efficiency of the proposed work.

15. Conclusion should concise the entire research work.

16. Shortcomings and future work of the proposed work should be provided.

Reviewer 2 Report

1.      In the abstract, the problem of the study is clearly defined but the authors have not included the limitations of the previous research. Along with this, the numerical findings, novelty, and future work must be included in the abstract.

2.      In the introduction section, the literature is included for the identified problem. However, the research gap and findings of the literature must be incorporated below the literature. In order to include literature in the introduction section, a sub-section must be included.

3.      In the introduction, the study needs to discuss the distinct issues of face recognition and highlight why face occlusion is treated as a key issue in this study of facial recognition. Also, address why Block Permutation Linear Regression Classification is considered in this study.

4.      The study discussed and presented the analysis of Block Permutation Linear Regression Classification for face Occlusion. However, the authors must create a new section above the conclusion to discuss the results briefly.

5.      A comparative analysis table must be included for discussing the previous literature with the proposed study for concluding the research gap and the novelty of the study.

6.      In the conclusion section, the different findings in the study need to be highlighted and also discuss how they are impactful on the face occlusion problems. Moreover, the novelty and future work of study also need to include.

Reviewer 3 Report

1.      The problem of the study is clearly defined in the abstract, but the authors have not should include the previous limitations of the research. In addition, the abstract must include the numerical findings, novelty, and future work.

2.      In the introduction section, the research gap in the literature needs to be included. A comparison of the previous study need to presented under literature section.

3.      The study discussed and presented the analysis of Block Permutation Linear Regression Classification for face Occlusion.

4.      The various findings in the study must be highlighted in the conclusion section, as well as how they impact the face occlusion problem. Furthermore, the novelty and future work of the study must be included.

5. The methodology needs to be presented in the form of pictorial representation.

Reviewer 4 Report

This paper shows a novel method to descirbe the feature of face to classification, it works good and has good result. The literature is good and the text described clearly. The experimets are abundant to support the methods.

Author Response

Thank you very much for your comments!

Round 2

Reviewer 1 Report

The authors have incorporated the suggestions given in the initial review. The quality of the paper is drastically improved and it posses potential interest for possible publication. But still few errors has to be rectified.

The image quality of figures 6, 9, 11 has to be improved. Poor resolution of the image would lead to readers disinterest in the paper.

Recent works to be cited and briefed for sound comparative of the proposed work.
